# Distributed Flexible Nonlinear Tensor Factorization

**Shandian Zhe**[§]**, Kai Zhang**[†]**, Pengyuan Wang**[‡]**, Kuang-chih Lee**[♯]**, Zenglin Xu**[♮]**,**
**Yuan Qi**[♭]**, Zoubin Gharamani**[⋆]

[§]Dept. Computer Science, Purdue University, [†]NEC Laboratories America, Princeton NJ,
[‡]Dept. Marketing, University of Georgia at Athens, [♯]Yahoo! Research,
[♮]Big Data Res. Center, School Comp. Sci. Eng., Univ. of Electr. Sci. & Tech. of China,
[♭]Ant Financial Service Group, Alibaba, [⋆]University of Cambridge
[§]`szhe@purdue.edu`, [†]`kzhang@nec-labs.com`, [‡]`pengyuan@uga.edu`,
[♯]`kclee@yahoo-inc.com`, [♮]`zlxu@uestc.edu.cn`,
[♭]`alanqi0@outlook.com`, [⋆]`zoubin@cam.ac.uk`

## Abstract

Tensor factorization is a powerful tool to analyse multi-way data. Recently proposed nonlinear factorization methods, although capable of capturing complex relationships, are computationally quite expensive and may suffer a severe learning bias in case of extreme data sparsity. Therefore, we propose a distributed, flexible nonlinear tensor factorization model, which avoids the expensive computations and structural restrictions of the Kronecker-product in the existing TGP formulations, allowing an arbitrary subset of tensorial entries to be selected for training. Meanwhile, we derive a tractable and tight variational evidence lower bound (ELBO) that enables highly decoupled, parallel computations and high-quality inference. Based on the new bound, we develop a distributed, key-value-free inference algorithm in the MAPREDUCE framework, which can fully exploit the memory cache mechanism in fast MAPREDUCE systems such as SPARK. Experiments demonstrate the advantages of our method over several state-of-the-art approaches, in terms of both predictive performance and computational efficiency.

## 1 Introduction

Tensors, or multidimensional arrays, are generalizations of matrices (from binary interactions) to high-order interactions between multiple entities. For example, we can extract a three-mode tensor (*user*, *advertisement*, *context*) from online advertising logs. To analyze tensor data, people usually turn to factorization approaches, which use a set of latent factors to represent each entity and model how the latent factors interact with each other to generate tensor elements. Classical tensor factorization models, including Tucker [18] and CANDECOMP/PARAFAC (CP) [5], assume multilinear interactions and hence are unable to capture more complex, nonlinear relationships. Recently, Xu et al. [19] proposed Infinite Tucker decomposition (InfTucker), which generalizes the Tucker model to infinite feature space using a Tensor-variate Gaussian process (TGP) and is hence more powerful in modeling intricate nonlinear interactions. However, InfTucker and its variants [22, 23] are computationally expensive, because the Kronecker product between the covariances of all the modes requires the TGP to model the entire tensor structure. In addition, they may suffer from the extreme sparsity of real-world tensor data, i.e., when the proportion of the nonzero entries is extremely low. As is often the case, most of the zero elements in real tensors are meaningless: they simply indicate missing or unobserved entries. Incorporating all of them in the training process may affect the factorization quality and lead to biased predictions.

To address these issues, we propose a distributed, flexible nonlinear tensor factorization model, which has several important advantages. First, it can capture highly nonlinear interactions in the tensor, and is flexible enough to incorporate arbitrary subset of (meaningful) tensor entries for the training. This is achieved by placing a Gaussian process prior over tensor entries, where the input is constructed by concatenating the latent factors from each mode and the intricate relationships

are captured by using the kernel function. By using such a construction, the covariance function is then free of the Kronecker-product structure, and as a result users can freely choose any subset of tensor elements for the training process and incorporate prior domain knowledge. For example, one can choose a combination of balanced zero and nonzero elements to overcome the learning bias. Second, the tight variational evidence lower bound (ELBO) we derived using functional derivatives and convex conjugates subsumes optimal variational posteriors, thus evades inefficient, sequential E-M updates and enables highly efficient, parallel computations as well as improved inference quality. Moreover, the new bound allows us to develop a distributed, gradient-based optimization algorithm. Finally, we develop a simple yet very efficient procedure to avoid the data shuffling operation, a major performance bottleneck in the (key-value) sorting procedure in MAPREDUCE. That is, rather than sending out key-value pairs, each mapper simply calculates and sends a global gradient vector without keys. This key-value-free procedure is general and can effectively prevent massive disk IOs and fully exploit the memory cache mechanism in fast MAPREDUCE systems, such as SPARK.

Evaluation using small real-world tensor data have fully demonstrated the superior prediction accuracy of our model in comparison with InfTucker and other state-of-the-art; on large tensors with millions of nonzero elements, our approach is significantly better than, or at least as good as two popular large-scale nonlinear factorization methods based on TGP: one uses hierarchical modeling to perform distributed infinite Tucker decomposition [22]; the other further enhances InfTucker by using Dirichlet process mixture prior over the latent factors and employs an online learning scheme [23]. Our method also outperforms GigaTensor [8], a typical large-scale CP factorization algorithm, by a large margin. In addition, our method achieves a faster training speed and enjoys almost linear speedup with respect to the number of computational nodes. We apply our model to CTR prediction for online advertising and achieves a significant, $20\%$ improvement over the popular logistic regression and linear SVM approaches (Section 4 of the supplementary material).

## 2   Background

We first introduce the background knowledge. For convenience, we will use the same notations in [19]. Specifically, we denote a $K$-mode tensor by $\mathcal{M} \in \mathbb{R}^{d_1 \times \ldots \times d_K}$, where the $k$-th mode is of dimension $d_k$. The tensor entry at location $\mathbf{i}$ ($\mathbf{i} = (i_1, \ldots, i_K)$) is denoted by $m_{\mathbf{i}}$. To introduce Tucker decomposition, we need to generalize matrix-matrix products to tensor-matrix products. Specifically, a tensor $\mathcal{W} \in \mathbb{R}^{r_1 \times \ldots \times r_K}$ can multiply with a matrix $\mathbf{U} \in \mathbb{R}^{s \times t}$ at mode $k$ when its dimension at mode-$k$ is consistent with the number of columns in $\mathbf{U}$, i.e., $r_k = t$. The product is a new tensor, with size $r_1 \times \ldots \times r_{k-1} \times s \times r_{k+1} \times \ldots \times r_K$. Each element is calculated by $(\mathcal{W} \times_k \mathbf{U})_{i_1 \ldots i_{k-1} j i_{k+1} \ldots i_K} = \sum_{i_k=1}^{r_k} w_{i_1 \ldots i_K} u_{j i_k}$.

The Tucker decomposition model uses a latent factor matrix $\mathbf{U}_k \in \mathbb{R}^{d_k \times r_k}$ in each mode $k$ and a core tensor $\mathcal{W} \in \mathbb{R}^{r_1 \times \ldots \times r_K}$ and assumes the whole tensor $\mathcal{M}$ is generated by $\mathcal{M} = \mathcal{W} \times_1 \mathbf{U}^{(1)} \times_2 \ldots \times_K \mathbf{U}^{(K)}$. Note that this is a multilinear function of $\mathcal{W}$ and $\{\mathbf{U}_1, \ldots, \mathbf{U}_K\}$. It can be further simplified by restricting $r_1 = r_2 = \ldots = r_K$ and the off-diagonal elements of $\mathcal{W}$ to be 0. In this case, the Tucker model becomes CANDECOMP/PARAFAC (CP).

The infinite Tucker decomposition (InfTucker) generalizes the Tucker model to infinite feature space via a tensor-variate Gaussian process (TGP) [19]. Specifically, in a probabilistic framework, we assign a standard normal prior over each element of the core tensor $\mathcal{W}$, and then marginalize out $\mathcal{W}$ to obtain the probability of the tensor given the latent factors:

$$p(\mathcal{M}|\mathbf{U}^{(1)}, \ldots, \mathbf{U}^{(K)}) = \mathcal{N}(\text{vec}(\mathcal{M}); \mathbf{0}, \Sigma^{(1)} \otimes \ldots \otimes \Sigma^{(K)}) \tag{1}$$

where $\text{vec}(\mathcal{M})$ is the vectorized whole tensor, $\Sigma^{(k)} = \mathbf{U}^{(k)}\mathbf{U}^{(k)^\top}$ and $\otimes$ is the Kronecker-product. Next, we apply the kernel trick to model nonlinear interactions between the latent factors: Each row $\mathbf{u}_t^k$ of the latent factors $\mathbf{U}^{(k)}$ is replaced by a nonlinear feature transformation $\phi(\mathbf{u}_t^k)$ and thus an equivalent nonlinear covariance matrix $\Sigma^{(k)} = k(\mathbf{U}^{(k)}, \mathbf{U}^{(k)})$ is used to replace $\mathbf{U}^{(k)}\mathbf{U}^{(k)^\top}$, where $k(\cdot, \cdot)$ is the covariance function. After the nonlinear feature mapping, the original Tucker decomposition is performed in an (unknown) infinite feature space. Further, since the covariance of $\text{vec}(\mathcal{M})$ is a function of the latent factors $\mathcal{U} = \{\mathbf{U}^{(1)}, \ldots, \mathbf{U}^{(K)}\}$, Equation (1) actually defines a Gaussian process (GP) on tensors, namely tensor-variate GP (TGP) [19], where the input are based on $\mathcal{U}$. Finally, we can use different noisy models $p(\mathcal{Y}|\mathcal{M})$ to sample the observed tensor $\mathcal{Y}$. For example, we can use Gaussian models and Probit models for continuous and binary observations, respectively.

## 3 Model

Despite being able to capture nonlinear interactions, InfTucker may suffer from the extreme sparsity issue in real-world tensor data sets. The reason is that its full covariance is a Kronecker-product between the covariances over all the modes—$\{\Sigma^{(1)}, \ldots, \Sigma^{(K)}\}$ (see Equation (1)). Each $\Sigma^{(k)}$ is of size $d_k \times d_k$ and the full covariance is of size $\prod_k d_k \times \prod_k d_k$. Thus TGP is projected onto the entire tensor with respect to the latent factors $\mathcal{U}$, including all zero and nonzero elements, rather than a (meaningful) subset of them. However, the real-world tensor data are usually extremely sparse, with a huge number of zero entries and a tiny portion of nonzero entries. On one hand, because most zero entries are meaningless—they are either missing or unobserved, using them can adversely affect the tensor factorization quality and lead to biased predictions; on the other hand, incorporating numerous zero entries into GP models will result in large covariance matrices and high computational costs. Zhe et al. [22, 23] proposed to improve the scalability by modeling subtensors instead, but the sampled subtensors can still be very sparse. Even worse, because they are typically of small dimensions (for efficiency considerations), it is often possible to encounter subtensors full of zeros. This may further incur numerical instabilities in model estimation.

To address these issues, we propose a flexible Gaussian process tensor factorization model. While inheriting the nonlinear modeling power, our model disposes of the Kronecker-product structure in the full covariance and can therefore select an arbitrary subset of tensor entries for training.

Specifically, given a tensor $\mathcal{M} \in \mathbb{R}^{d_1 \times \cdots \times d_K}$, for each tensor entry $m_{\mathbf{i}}$ ($\mathbf{i} = (i_1, \ldots, i_K)$), we construct an input $\mathbf{x}_i$ by concatenating the corresponding latent factors from all the modes: $\mathbf{x}_{\mathbf{i}} = [\mathbf{u}_{i_1}^{(1)}, \ldots, \mathbf{u}_{i_K}^{(K)}]$, where $\mathbf{u}_{i_k}^{(k)}$ is the $i_k$-th row in the latent factor matrix $\mathbf{U}^{(k)}$ for mode $k$. We assume that there is an underlying function $f : \mathbb{R}^{\sum_{j=1}^{K} d_j} \to \mathbb{R}$ such that $m_{\mathbf{i}} = f(\mathbf{x}_{\mathbf{i}}) = f([\mathbf{u}_{i_1}^{(1)}, \ldots, \mathbf{u}_{i_K}^{(K)}])$. This function is unknown and can be complex and nonlinear. To learn the function, we assign a Gaussian process prior over $f$: for any set of tensor entries $S = \{\mathbf{i}_1, \ldots, \mathbf{i}_N\}$, the function values $\mathbf{f}_S = \{f(\mathbf{x}_{\mathbf{i}_1}), \ldots, f(\mathbf{x}_{\mathbf{i}_N})\}$ are distributed according to a multivariate Gaussian distribution with mean $\mathbf{0}$ and covariance determined by $\mathbf{X}_S = \{\mathbf{x}_{\mathbf{i}_1}, \ldots, \mathbf{x}_{\mathbf{i}_N}\}$:

$$p(\mathbf{f}_S|\mathcal{U}) = \mathcal{N}(\mathbf{f}_S|\mathbf{0}, k(\mathbf{X}_S, \mathbf{X}_S))$$

where $k(\cdot, \cdot)$ is a (nonlinear) covariance function.

Because $k(\mathbf{x}_{\mathbf{i}}, \mathbf{x}_{\mathbf{j}}) = k([\mathbf{u}_{i_1}^{(1)}, \ldots, \mathbf{u}_{i_K}^{(K)}], [\mathbf{u}_{j_1}^{(1)}, \ldots, \mathbf{u}_{j_K}^{(K)}])$, there is no Kronecker-product structure constraint and so any subset of tensor entries can be selected for training. To prevent the learning process to be biased toward zero, we can use a set of entries with balanced zeros and nonzeros; furthermore, useful domain knowledge can also be incorporated to select meaningful entries for training. Note, however, that if we still use all the tensor entries and intensionally impose the Kronecker-product structure in the full covariance, our model is reduced to InfTucker. Therefore, from the modeling perspective, the proposed model is more general.

We further assign a standard normal prior over the latent factors $\mathcal{U}$. Given the selected tensor entries $\mathbf{m} = [m_{\mathbf{i}_1}, \ldots, m_{\mathbf{i}_N}]$, the observed entries $\mathbf{y} = [y_{\mathbf{i}_1}, \ldots, y_{\mathbf{i}_N}]$ are sampled from a noise model $p(\mathbf{y}|\mathbf{m})$. In this paper, we deal with both continuous and binary observations. For continuous data, we use the Gaussian model, $p(\mathbf{y}|\mathbf{m}) = \mathcal{N}(\mathbf{y}|\mathbf{m}, \beta^{-1}\mathbf{I})$ and the joint probability is

$$p(\mathbf{y}, \mathbf{m}, \mathcal{U}) = \prod_{t=1}^{K} \mathcal{N}(\text{vec}(\mathbf{U}^{(t)})|\mathbf{0}, \mathbf{I})\mathcal{N}(\mathbf{m}|\mathbf{0}, k(\mathbf{X}_S, \mathbf{X}_S))\mathcal{N}(\mathbf{y}|\mathbf{m}, \beta^{-1}\mathbf{I}) \tag{2}$$

where $S = [\mathbf{i}_1, \ldots, \mathbf{i}_N]$. For binary data, we use the Probit model in the following manner. We first introduce augmented variables $\mathbf{z} = [z_1, \ldots, z_N]$ and then decompose the Probit model into $p(z_j|m_{\mathbf{i}_j}) = \mathcal{N}(z_j|m_{\mathbf{i}_j}, 1)$ and $p(y_{\mathbf{i}_j}|z_j) = \mathbb{1}(y_{\mathbf{i}_j} = 0)\mathbb{1}(z_j \leq 0) + \mathbb{1}(y_{\mathbf{i}_j} = 1)\mathbb{1}(z_j > 0)$ where $\mathbb{1}(\cdot)$ is the indicator function. Then the joint probability is

$$p(\mathbf{y}, \mathbf{z}, \mathbf{m}, \mathcal{U}) = \prod_{t=1}^{K} \mathcal{N}(\text{vec}(\mathbf{U}^{(t)})|\mathbf{0}, \mathbf{I})\mathcal{N}(\mathbf{m}|\mathbf{0}, k(\mathbf{X}_S, \mathbf{X}_S))\mathcal{N}(\mathbf{z}|\mathbf{m}, \mathbf{I})$$
$$\cdot \prod_j \mathbb{1}(y_{\mathbf{i}_j} = 0)\mathbb{1}(z_j \leq 0) + \mathbb{1}(y_{\mathbf{i}_j} = 1)\mathbb{1}(z_j > 0). \tag{3}$$

## 4 Distributed Variational Inference

Real-world tensors often comprise a large number of entries, say, millions of non-zeros and billions of zeros, making exact inference of the proposed model totally intractable. This motives us to develop a distributed variational inference algorithm, presented as follows.

## 4.1 Tractable Variational Evidence Lower Bound

Since the GP covariance term — $k(\mathbf{X}_S, \mathbf{X}_S)$ (see Equations (2) and (3)) intertwines all the latent factors, exact inference in parallel is quite difficult. Therefore, we first derive a tractable variational evidence lower bound (ELBO), following the sparse Gaussian process framework by Titsias [17]. The key idea is to introduce a small set of inducing points $\mathbf{B} = \{\mathbf{b}_1, \ldots, \mathbf{b}_p\}$ and latent targets $\mathbf{v} = \{v_1, \ldots, v_p\}$ ($p \ll N$). Then we augment the original model with a joint multivariate Gaussian distribution of the latent tensor entries $\mathbf{m}$ and targets $\mathbf{v}$, $p(\mathbf{m}, \mathbf{v}|\mathcal{U}, \mathbf{B}) = \mathcal{N}([\mathbf{m}, \mathbf{v}]^\top|[\mathbf{0}, \mathbf{0}]^\top, [\mathbf{K}_{SS}, \mathbf{K}_{SB}; \mathbf{K}_{BS}, \mathbf{K}_{BB}])$ where $\mathbf{K}_{SS} = k(\mathbf{X}_S, \mathbf{X}_S)$, $\mathbf{K}_{BB} = k(\mathbf{B}, \mathbf{B})$, $\mathbf{K}_{SB} = k(\mathbf{X}_S, \mathbf{B})$ and $\mathbf{K}_{BS} = k(\mathbf{B}, \mathbf{X}_S)$. We use Jensen's inequality and conditional Gaussian distributions to construct the ELBO. Using a very similar derivation to [17], we can obtain a tractable ELBO for our model on continuous data, $\log\big(p(\mathbf{y}, \mathcal{U}|\mathbf{B})\big) \geq L_1\big(\mathcal{U}, \mathbf{B}, q(\mathbf{v})\big)$, where

$$L_1\big(\mathcal{U}, \mathbf{B}, q(\mathbf{v})\big) = \log(p(\mathcal{U})) + \int q(\mathbf{v})\log\frac{p(\mathbf{v}|\mathbf{B})}{q(\mathbf{v})}\mathrm{d}\mathbf{v} + \sum_j \int q(\mathbf{v}) F_{\mathbf{v}}(y_{\mathbf{i}_j}, \beta)\mathrm{d}\mathbf{v}. \quad (4)$$

Here $p(\mathbf{v}|\mathbf{B}) = \mathcal{N}(\mathbf{v}|\mathbf{0}, K_{BB})$, $q(\mathbf{v})$ is the variational posterior for the latent targets $\mathbf{v}$ and $F_{\mathbf{v}}(\cdot_j, *) = \int \log\big(\mathcal{N}(\cdot_j|m_{\mathbf{i}_j}, *)\big)\mathcal{N}(m_{\mathbf{i}_j}|\mu_j, \sigma_j^2)\mathrm{d}m_{\mathbf{i}_j}$, where $\mu_j = k(\mathbf{x}_{\mathbf{i}_j}, \mathbf{B})\mathbf{K}_{BB}^{-1}\mathbf{v}$ and $\sigma_j^2 = \mathbf{\Sigma}(j, j) = k(\mathbf{x}_{\mathbf{i}_j}, \mathbf{x}_{\mathbf{i}_j}) - k(\mathbf{x}_{\mathbf{i}_j}, \mathbf{B})\mathbf{K}_{BB}^{-1}k(\mathbf{B}, \mathbf{x}_{\mathbf{i}_j})$. Note that $L_1$ is decomposed into a summation of terms involving individual tensor entries $\mathbf{i}_j (1 \leq j \leq N)$. The additive form enables us to distribute the computation across multiple computers.

For binary data, we introduce a variational posterior $q(\mathbf{z})$ and make the mean-field assumption that $q(\mathbf{z}) = \prod_j q(z_j)$. Following a similar derivation to the continuous case, we can obtain a tractable ELBO for binary data, $\log\big(p(\mathbf{y}, \mathcal{U}|\mathbf{B})\big) \geq L_2\big(\mathcal{U}, \mathbf{B}, q(\mathbf{v}), q(\mathbf{z})\big)$, where

$$L_2\big(\mathcal{U}, \mathbf{B}, q(\mathbf{v}), q(\mathbf{z})\big) = \log(p(\mathcal{U})) + \int q(\mathbf{v})\log\big(\frac{p(\mathbf{v}|\mathbf{B})}{q(\mathbf{v})}\big)\mathrm{d}\mathbf{v} + \sum_j q(z_j)\log\big(\frac{p(y_{\mathbf{i}_j}|z_j)}{q(z_j)}\big)$$

$$+ \sum_j \int q(\mathbf{v}) \int q(z_j) F_{\mathbf{v}}(z_j, 1)\mathrm{d}z_j\mathrm{d}\mathbf{v}. \quad (5)$$

One can simply use the standard Expectation-maximization (EM) framework to optimize (4) and (5) for model inference, i.e., the E step updates the variational posteriors $\{q(\mathbf{v}), q(\mathbf{z})\}$ and the M step updates the latent factors $\mathcal{U}$, the inducing points $\mathbf{B}$ and the kernel parameters. However, the sequential E-M updates can not fully exploit the paralleling computing resources. Due to the strong dependencies between the E step and the M step, the sequential E-M updates may take a large number of iterations to converge. Things become worse for binary case: in the E step, the updates of $q(\mathbf{v})$ and $q(\mathbf{z})$ are also dependent on each other, making a parallel inference even less efficient.

## 4.2 Tight and Parallelizable Variational Evidence Lower Bound

In this section, we further derive tight(er) ELBOs that subsume the optimal variational posteriors for $q(\mathbf{v})$ and $q(\mathbf{z})$. Thereby we can avoid the sequential E-M updates to perform decoupled, highly efficient parallel inference. Moreover, the inference quality is very likely to be improved using tighter bounds. Due to the space limit, we only present key ideas and results here; detailed discussions are given in Section 1 and 2 of the supplementary material.

**Tight ELBO for continuous tensors.** We take functional derivative of $L_1$ with respect to $q(\mathbf{v})$ in (4). By setting the derivative to zero, we obtain the optimal $q(\mathbf{v})$ (which is a Gaussian distribution) and then substitute it into $L_1$, manipulating the terms, we achieve the following tighter ELBO.

**Theorem 4.1.** *For continuous data, we have*

$$\log\big(p(\mathbf{y}, \mathcal{U}|\mathbf{B})\big) \geq L_1^*(\mathcal{U}, \mathbf{B}) = \frac{1}{2}\log|\mathbf{K}_{BB}| - \frac{1}{2}\log|\mathbf{K}_{BB} + \beta\mathbf{A}_1| - \frac{1}{2}\beta a_2 - \frac{1}{2}\beta a_3$$

$$+ \frac{\beta}{2}\mathrm{tr}(\mathbf{K}_{BB}^{-1}\mathbf{A}_1) - \frac{1}{2}\sum_{k=1}^{K}\|\mathbf{U}^{(k)}\|_F^2 + \frac{1}{2}\beta^2\mathbf{a}_4^\top(\mathbf{K}_{BB} + \beta\mathbf{A}_1)^{-1}\mathbf{a}_4 + \frac{N}{2}\log(\frac{\beta}{2\pi}), \quad (6)$$

*where $\|\cdot\|_F$ is Frobenius norm, and*

$$\mathbf{A}_1 = \sum_j k(\mathbf{B}, \mathbf{x}_{\mathbf{i}_j})k(\mathbf{x}_{\mathbf{i}_j}, \mathbf{B}), \quad a_2 = \sum_j y_{\mathbf{i}_j}^2, \quad a_3 = \sum_j k(\mathbf{x}_{\mathbf{i}_j}, \mathbf{x}_{\mathbf{i}_j}), \quad \mathbf{a}_4 = \sum_j k(\mathbf{B}, \mathbf{x}_{\mathbf{i}_j})y_{\mathbf{i}_j}.$$

**Tight ELBO for binary tensors.** The binary case is more difficult because $q(\mathbf{v})$ and $q(\mathbf{z})$ are coupled together (see (5)). We use the following steps: we first fix $q(\mathbf{z})$ and plug the optimal $q(\mathbf{v})$ in the same way as the continuous case. Then we obtain an intermediate ELBO $\hat{L}_2$ that only contains $q(\mathbf{z})$. However, a quadratic term in $\hat{L}_2$ , $\frac{1}{2}(\mathbf{K}_{BS}\langle\mathbf{z}\rangle)^\top(\mathbf{K}_{BB} + \mathbf{A}_1)^{-1}(\mathbf{K}_{BS}\langle\mathbf{z}\rangle)$, intertwines all $\{q(z_j)\}_j$ in $\hat{L}_2$, making it infeasible to analytically derive or parallelly compute the optimal $\{q(z_j)\}_j$. To overcome this difficulty, we use the convex conjugate of the quadratic term, and introduce a variational parameter $\boldsymbol{\lambda}$ to decouple the dependences between $\{q(z_j)\}_j$. After that, we are able to derive the optimal $\{q(z_j)\}_j$ using functional derivatives and to obtain the following tight ELBO.

**Theorem 4.2.** *For binary data, we have*

$$\log\big(p(\mathbf{y},\mathcal{U}|\mathbf{B})\big) \geq L_2^*(\mathcal{U},\mathbf{B},\boldsymbol{\lambda}) = \frac{1}{2}\log|\mathbf{K}_{BB}| - \frac{1}{2}\log|\mathbf{K}_{BB} + \mathbf{A}_1| - \frac{1}{2}a_3$$

$$+ \sum_j \log\big(\Phi((2y_{\mathbf{i}_j} - 1)\boldsymbol{\lambda}^\top k(\mathbf{B},\mathbf{x}_{\mathbf{i}_j}))\big) - \frac{1}{2}\boldsymbol{\lambda}^\top\mathbf{K}_{BB}\boldsymbol{\lambda} + \frac{1}{2}\mathrm{tr}(\mathbf{K}_{BB}^{-1}\mathbf{A}_1) - \frac{1}{2}\sum_{k=1}^K\|\mathbf{U}^{(k)}\|_F^2 \quad (7)$$

*where $\Phi(\cdot)$ is the cumulative distribution function of the standard Gaussian.*

As we can see, due to the additive forms of the terms in $L_1^*$ and $L_2^*$, such as $\mathbf{A}_1$, $a_2$, $a_3$ and $\mathbf{a}_4$, the computation of the tight ELBOs and their gradients can be efficiently performed in parallel.

## 4.3 Distributed Inference on Tight Bound

### 4.3.1 Distributed Gradient-based Optimization

Given the tighter ELBOs in (6) and (7), we develop a distributed algorithm to optimize the latent factors $\mathcal{U}$, the inducing points $\mathbf{B}$, the variational parameters $\boldsymbol{\lambda}$ (for binary data) and the kernel parameters. We distribute the computations over multiple computational nodes (MAP step) and then collect the results to calculate the ELBO and its gradient (REDUCE step). A standard routine, such as gradient descent and L-BFGS, is then used to solve the optimization problem.

For binary data, we further find that $\boldsymbol{\lambda}$ can be updated with a simple fixed point iteration:

$$\boldsymbol{\lambda}^{(t+1)} = (\mathbf{K}_{BB} + \mathbf{A}_1)^{-1}(\mathbf{A}_1\boldsymbol{\lambda}^{(t)} + \mathbf{a}_5) \quad (8)$$

where $\mathbf{a}_5 = \sum_j k(\mathbf{B},\mathbf{x}_{\mathbf{i}_j})(2y_{\mathbf{i}_j} - 1)\frac{\mathcal{N}\big(k(\mathbf{B},\mathbf{x}_{\mathbf{i}_j})^\top\boldsymbol{\lambda}^{(t)}|0,1\big)}{\Phi\big((2y_{\mathbf{i}_j}-1)k(\mathbf{B},\mathbf{x}_{\mathbf{i}_j})^\top\boldsymbol{\lambda}^{(t)}\big)}$.

Apparently, the updating can be efficiently performed in parallel (due to the additive structure of $\mathbf{A}_1$ and $\mathbf{a}_5$). Moreover, the convergence is guaranteed by the following lemma. The proof is given in Section 3 of the supplementary material.

**Lemma 4.3.** *Given $\mathcal{U}$ and $\mathbf{B}$, we have $L_2^*(\mathcal{U},\mathbf{B},\boldsymbol{\lambda}^{t+1}) \geq L_2^*(\mathcal{U},\mathbf{B},\boldsymbol{\lambda}^t)$ and the fixed point iteration* (8) *always converges.*

To use the fixed point iteration, before we calculate the gradients with respect to $\mathcal{U}$ and $\mathbf{B}$, we first optimize $\boldsymbol{\lambda}$ via (8) in an inner loop. In the outer control, we then employ gradient descent or L-BFGS to optimize $\mathcal{U}$ and $\mathbf{B}$. This will lead to an even tighter bound for our model: $L_2^{**}(\mathcal{U},\mathbf{B}) = \max_{\boldsymbol{\lambda}} L_2^*(\mathcal{U},\mathbf{B},\boldsymbol{\lambda}) = \max_{q(\mathbf{v}),q(\mathbf{z})} L_2(\mathcal{U},\mathbf{B},q(\mathbf{v}),q(\mathbf{z}))$. Empirically, this converges must faster than feeding the optimization algorithms with $\partial\lambda$, $\partial\mathcal{U}$ and $\partial\mathbf{B}$ altogether, especially for large data.

### 4.3.2 Key-Value-Free MAPREDUCE

We now present the detailed design of MAPREDUCE procedures to fulfill our distributed inference. Basically, we first allocate a set of tensor entries $S_t$ on each MAPPER $t$ such that the corresponding components of the ELBO and the gradients are calculated; then the REDUCER aggregates local results from each MAPPER to obtain the integrated, global ELBO and gradient.

We first consider the standard (key-value) design. For brevity, we take the gradient computation for the latent factors as an example. For each tensor entry $\mathbf{i}$ on a MAPPER, we calculate the corresponding gradients $\{\partial\mathbf{u}_{i_1}^{(1)},\ldots\partial\mathbf{u}_{i_K}^{(K)}\}$ and then send out the key-value pairs $\{(k,i_k) \rightarrow \partial\mathbf{u}_{i_k}^{(k)}\}_k$, where the key indicates the mode and the index of the latent factors. The REDUCER aggregates gradients with the same key to recover the full gradient with respect to each latent factor.

Although the (key-value) MAPREDUCE has been successfully applied in numerous applications, it relies on an expensive data shuffling operation: the REDUCE step has to sort the MAPPERS' output by the keys before aggregation. Since the sorting is usually performed on disk due to significant data size, intensive disk I/Os and network communications will become serious computational overheads. To overcome this deficiency, we devise a key-value-free MAP-REDUCE scheme to avoid on-disk data shuffling operations. Specifically, on each MAPPER, a complete gradient vector is maintained for all the parameters, including $\mathcal{U}$, $\mathbf{B}$ and the kernel parameters; however, only relevant components of the gradient, as specified by the tensor entries allocated to this MAPPER, will be updated. After updates, each MAPPER will then send out the full gradient vector, and the REDUCER will simply sum them up together to obtain a global gradient vector without having to perform any extra data sorting. Note that a similar procedure can also be used to perform the fixed point iteration for $\boldsymbol{\lambda}$ (in binary tensors).

Efficient MAPREDUCE systems, such as SPARK [21], can fully optimize the non-shuffling MAP and REDUCE, where most of the data are buffered in memory and disk I/Os are circumvented to the utmost; by contrast, the performance with data shuffling degrades severely [3]. This is verified in our evaluations: on a small tensor of size $100 \times 100 \times 100$, our key-value-free MAPREDUCE gains 30 times speed acceleration over the traditional key-value process. Therefore, our algorithm can fully exploit the memory-cache mechanism to achieve fast inference.

### 4.4 Algorithm Complexity

Suppose we use $N$ tensor entries for training, with $p$ inducing points and $T$ MAPPER, the time complexity for each MAPPER node is $O(\frac{1}{T}p^2 N)$. Since $p \ll N$ is a fixed constant ($p = 100$ in our experiments), the time complexity is linear in the number of tensor entries. The space complexity for each MAPPER node is $O(\sum_{j=1}^{K} m_j r_j + p^2 + \frac{N}{T}K)$, in order to store the latent factors, their gradients, the covariance matrix on inducing points, and the indices of the latent factors for each tensor entry. Again, the space complexity is linear in the number of tensor entries. In comparison, InfTucker utilizes the Kronecker-product properties to calculate the gradients and has to perform eigenvalue decomposition of the covariance matrices in each tensor mode. Therefor it has a higher time and space complexity (see [19] for details) and is not scalable to large dimensions.

## 5 Related work

Classical tensor factorization models include Tucker [18] and CP [5], based on which there are many excellent works [2, 16, 6, 20, 14, 7, 13, 8, 1]. Despite the wide-spread success, their underlying multilinear factorization structures prevent them from capturing more complex, nonlinear relationship in real-world applications. Infinite Tucker decomposition [19], and its distributed or online extensions [22, 23] overcome this limitation by modeling tensors or subtensors via tensor-variate Gaussian processes (TGP). However, these methods may suffer from the extreme sparsity in real-world tensors due to the Kronecker-product structure in TGP formulations. Our model further address this issue by eliminating the Kronecker-product restriction, and can model an arbitrary subset of tensor entries. In theory, all such nonlinear factorization models belong to the family of random function prior models [11] for exchangeable multidimensional arrays.

Our distributed variational inference algorithm is based on sparse GP [12], an efficient approximation framework to scale up GP models. Sparse GP uses a small set of inducing points to break the dependency between random function values. Recently, Titsias [17] proposed a variational learning framework for sparse GP, based on which Gal et al. [4] derived a tight variational lower bound for distributed inference of GP regression and GPLVM [10]. The derivation of the tight ELBO in our model for continuous tensors is similar to [4]; however, the gradient calculation is substantially different, because the input to our GP factorization model is the concatenation of the latent factors. Many tensor entries may partly share the same latent factors, causing a large amount of key-value pair to be sent during the distributed gradient calculation. This will incur an expensive data shuffling procedure that takes place on disk. To improve the computational efficiency, we develop a non-key-value MAP-REDUCE to avoid data shuffling and fully exploit the memory-cache mechanism in efficient MAPREDUCE systems. This strategy is also applicable to other MAP-REDUCE based learning algorithms. In addition to continuous data, we also develop a tight ELBO for binary data on optimal variational posteriors. By introducing $p$ extra variational parameters with convex conjugates ($p$ is the number of inducing points), our inference can be performed efficiently in a distributed manner, which avoids explicit optimization on a large number of variational posteriors for the latent tensor entries and inducing targets. Our method can also be useful for GP classification problem.

# 6 Experiments

## 6.1 Evaluation on Small Tensor Data

For evaluation, we first compared our method with various existing tensor factorization methods. To this end, we used four small real datasets where all methods are computationally feasible: (1) *Alog*, a real-valued tensor of size $200 \times 100 \times 200$, representing a three-way interaction (user, action, resource) in a file access log. It contains $0.33\%$ nonzero entries.(2) *AdClick*, a real-valued tensor of size $80 \times 100 \times 100$, describing (user, publisher, advertisement) clicks for online advertising. It contains $2.39\%$ nonzero entries. (3) *Enron*, a binary tensor depicting the three-way relationship (sender, receiver, time) in emails. It contains $203 \times 203 \times 200$ elements, of which $0.01\%$ are nonzero. (4) *NellSmall*, a binary tensor of size $295 \times 170 \times 94$, depicting the knowledge predicates (entity, relationship, entity). The data set contains $0.05\%$ nonzero elements.

We compared with CP, nonnegative CP (NN-CP) [15], high order SVD (HOSVD) [9], Tucker, infinite Tucker (InfTucker) [19] and its extension (InfTuckerEx) which uses the Dirichlet process mixture (DPM) prior to model latent clusters and local TGP to perform scalable, online factorization [23]. Note that InfTucker and InfTuckerEx are nonlinear factorization approaches.

For testing, we used the same setting as in [23]. All the methods were evaluated via a 5-fold cross validation. The nonzero entries were randomly split into 5 folds; 4 folds were used for training and the remaining non-zero entries and $0.1\%$ zero entries were used for testing so that the number of non-zero entries is comparable to the number of zero entries. In doing so, zero and nonzero entries are treated equally important in testing, and the evaluation will not be dominated by large portion of zeros. For InfTucker and InfTuckerEx, we performed extra cross-validations to select the kernel form (*e.g.,* RBF, ARD and Matern kernels) and the kernel parameters. For InfTuckerEx, we randomly sampled subtensors and tuned the learning rate following [23]. For our model, the number of inducing points was set to 100, and we used a balanced training set generated as follows: in addition to nonzero entries, we randomly sampled the same number of zero entries and made sure that they would not overlap with the testing zero elements.

Our model used ARD kernel and the kernel parameters were estimated jointly with the latent factors. We implemented our distributed inference algorithm with two optimization frameworks, gradient descent and L-BFGS (denoted by Ours-GD and Ours-LBFGS respectively). For a comprehensive evaluation, we also examined CP on balanced training entries generated in the same way as our model, denoted by CP-2. The mean squared error (MSE) is used to evaluate predictive performance on *Alog* and *Click* and area-under-curve (AUC) on *Enron* and *NellSmall*. The averaged results from the 5-fold cross validation are reported.

Our model achieves a higher prediction accuracy than InfTucker, and a better or comparable accuracy than InfTuckerEx (see Figure 1). A $t$-test shows that our model outperforms InfTucker significantly ($p < 0.05$) in almost all situations. Although InfTuckerEx uses the DPM prior to improve factorization, our model still obtains significantly better predictions on *Alog* and *AdClick* and comparable or better performance on *Enron* and *NellSmall*. This might be attributed to the flexibility of our model in using balanced training entries to prevent the learning bias (toward numerous zeros). Similar improvements can be observed from CP to CP-2. Finally, our model outperforms all the remaining methods, demonstrating the advantage of our nonlinear factorization approach.

## 6.2 Scalability Analysis

To examine the scalability of the proposed distributed inference algorithm, we used the following large real-world datasets: (1) ACC, A real-valued tensor describing three-way interactions (user, action, resource) in a code repository management system [23]. The tensor is of size $3K \times 150 \times 30K$, where $0.009\%$ are nonzero. (2) DBLP: a binary tensor depicting a three-way bibliography relationship (author, conference, keyword) [23]. The tensor was extracted from DBLP database and contains $10K \times 200 \times 10K$ elements, where $0.001\%$ are nonzero entries. (3) NELL: a binary tensor representing the knowledge predicates, in the form of (entity, entity, relationship) [22]. The tensor size is $20K \times 12.3K \times 280$ and $0.0001\%$ are nonzero.

The scalability of our distributed inference algorithm was examined with regard to the number of machines on ACC dataset. The number of latent factors was set to 3. We ran our algorithm using the gradient descent. The results are shown in Figure 2(a). The Y-axis shows the reciprocal of the running time multiplied by a constant—which corresponds to the running speed. As we can see, the speed of our algorithm scales up linearly to the number of machines.

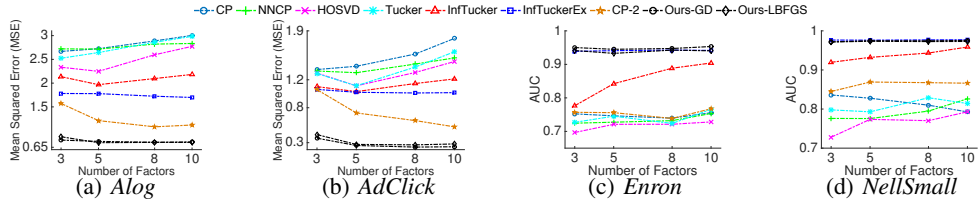

Figure 1: The prediction results on small datasets. The results are averaged over 5 runs.

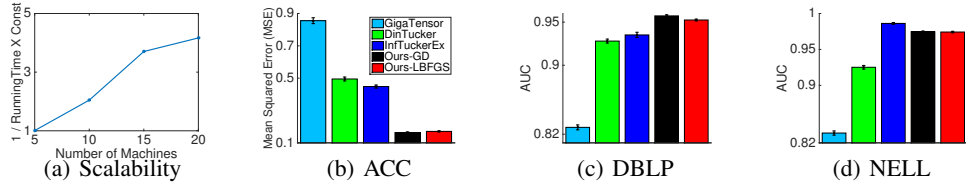

Figure 2: Prediction accuracy (averaged on 50 test datasets) on large tensor data and the scalability.

## 6.3 Evaluation on Large Tensor Data

We then compared our approach with three state-of-the-art large-scale tensor factorization methods: GigaTensor [8], Distributed infinite Tucker decomposition (DinTucker) [22], and InfTuckerEx [23]. Both GigaTensor and DinTucker are developed on HADOOP, while InfTuckerEx uses online inference. Our model was implemented on SPARK. We ran Gigatensor, DinTucker and our approach on a large YARN cluster and InfTuckerEx on a single computer.

We set the number of latent factors to 3 for ACC and DBLP data set, and 5 for NELL data set. Following the settings in [23, 22], we randomly chose 80% of nonzero entries for training, and then sampled 50 test data sets from the remaining entries. For ACC and DBLP, each test data set comprises 200 nonzero elements and 1,800 zero elements; for NELL, each test data set contains 200 nonzero elements and 2,000 zero elements. The running of GigaTensor was based on the default settings of the software package. For DinTucker and InfTuckerEx, we randomly sampled subtensors for distributed or online inference. The parameters, including the number and size of the subtensors and the learning rate, were selected in the same way as [23]. The kernel form and parameters were chosen by a cross-validation on the training tensor. For our model, we used the same setting as in the small data. We set 50 MAPPERS for GigaTensor, DinTucker and our model.

Figure 2(b)-(d) shows the predictive performance of all the methods. We observe that our approach consistently outperforms GigaTensor and DinTucker on all the three datasets; our approach outperforms InfTuckerEx on ACC and DBLP and is slightly worse than InfTuckerEx on NELL. Note again that InfTuckerEx uses DPM prior to enhance the factorization while our model doesn't; finally, all the nonlinear factorization methods outperform GigaTensor, a distributed CP factorization algorithm by a large margin, confirming the advantages of nonlinear factorizations on large data. In terms of speed, our algorithm is much faster than GigaTensor and DinTucker. For example, on DBLP dataset, the average per-iteration running time were 1.45, 15.4 and 20.5 minutes for our model, GigaTensor and DinTucker, respectively. This is not surprising, because (1) our model uses the data sparsity and can exclude numerous, meaningless zero elements from training; (2) our algorithm is based on SPARK, a more efficient MAPREDUCE system than HADOOP; (3) our algorithm gets rid of data shuffling and can fully exploit the memory-cache mechanism of SPARK.

## 7 Conclusion

In this paper, we have proposed a novel flexible GP tensor factorization model. In addition, we have derived a tight ELBO for both continuous and binary problems, based on which we further developed an efficient distributed variational inference algorithm in MAPREDUCE framework.

## Acknowledgement

Dr. Zenglin Xu was supported by a grant from NSF China under No. 61572111. We thank IBM T.J. Watson Research Center for providing one dataset. We also thank Jiasen Yang for proofreading this paper.

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
