[Supplementary Material]

**Supplementary Material**

 In this extra material, we provide the details about the derivation of the tight variational evidence lower
 bound of our proposed GP factorization model (Section 1) as well as its gradient calculation (Section
 2). Moreover, we give the convergence proof of the fixed point iteration used in our distributed
 inference algorithm for binary tensor (Section 3). Finally, we show the experimental results of our
 model's application on real-world click-through-rate prediction problem (Section 4).

## 7 1 Tight Variational Evidence Lower Bound

 The naive variational evidence lower bound (ELBO) derived from the sparse Gaussian process
 framework (see Section 4.1 of the main paper) is given by

$$L_1(\mathcal{U}, \mathbf{B}, q(\mathbf{v})) = \log(p(\mathcal{U})) + \int q(\mathbf{v}) \log \frac{p(\mathbf{v}|\mathbf{B})}{q(\mathbf{v})} \mathrm{d}\mathbf{v} + \sum_j \int q(\mathbf{v}) F_{\mathbf{v}}(y_{\mathbf{i}_j}, \beta) \mathrm{d}\mathbf{v} \qquad (1)$$

 for continuous tensor and

$$L_2(\mathcal{U}, \mathbf{B}, q(\mathbf{v}), q(\mathbf{z})) = \log(p(\mathcal{U})) + \int q(\mathbf{v}) \log(\frac{p(\mathbf{v}|\mathbf{B})}{q(\mathbf{v})}) \mathrm{d}\mathbf{v} + \sum_j q(z_j) \log(\frac{p(y_{\mathbf{i}_j}|z_j)}{q(z_j)})$$

$$+ \sum_j \int q(\mathbf{v}) \int q(z_j) F_{\mathbf{v}}(z_j, 1) \mathrm{d}z_j \mathrm{d}\mathbf{v} \qquad (2)$$

 for binary tensor, where $F_{\mathbf{v}}(\cdot_j, *) = \int \log \left( \mathcal{N}(\cdot_j | m_{\mathbf{i}_j}, *) \right) \mathcal{N}(m_{\mathbf{i}_j} | \mu_j, \sigma_j^2) \mathrm{d}m_{\mathbf{i}_j}$ and $p(\mathbf{v}|\mathbf{B}) =$
 $\mathcal{N}(\mathbf{v}|\mathbf{0}, K_{BB})$. Our goal is to further obtain a tight ELBO that subsumes the optimal variational
 posterior (i.e., $q(\mathbf{v})$ and $q(\mathbf{z})$) so as to prevent the sequential E-M procedure for efficient parallel
 training and to improve the inference quality.

### 15 1.1 Continuous Tensor

 First, let us consider the continuous data. Given $\mathcal{U}$ and $\mathbf{B}$, we use functional derivatives (Bishop,
 2006) to calculate the optimal $q(\mathbf{v})$. The functional derivative of $L_1$ with respect to $q(\mathbf{v})$ is given by

$$\frac{\delta L_1(q)}{\delta q(\mathbf{v})} = \log \frac{p(\mathbf{v}|\mathbf{B})}{q(\mathbf{v})} - 1 + \sum_j F_{\mathbf{v}}(y_{\mathbf{i}_j}, \beta).$$

 Because $q(\mathbf{v})$ is a probability density function, we use Lagrange multipliers to impose the constraint
 and obtain the optimal $q(\mathbf{v})$ by solving

$$\frac{\delta \left( L_1(q) + \lambda(\int q(\mathbf{v})\mathrm{d}\mathbf{v} - 1) \right)}{\delta q(\mathbf{v})} = 0,$$

$$\frac{\partial \left( L_1(q) + \lambda(\int q(\mathbf{v})\mathrm{d}\mathbf{v} - 1) \right)}{\partial \lambda} = 0.$$

 Though simple algebraic manipulations, we can obtain the optimal $q(\mathbf{v})$ to be the following form

$$q^*(\mathbf{v}) = \mathcal{N}(\mathbf{v}|\boldsymbol{\mu}, \boldsymbol{\Lambda}),$$

 where $\boldsymbol{\mu} = \beta \mathbf{K}_{BB}(\mathbf{K}_{BB} + \beta \mathbf{K}_{BS}\mathbf{K}_{SB})^{-1}\mathbf{K}_{BS}\mathbf{y}$ and $\boldsymbol{\Lambda} = \mathbf{K}_{BB}(\mathbf{K}_{BB} + \beta \mathbf{K}_{BS}\mathbf{K}_{SB})^{-1}\mathbf{K}_{BB}$.
 Now substituting $q(\mathbf{v})$ in $L_1$ with $\mathcal{N}(\mathbf{v}|\boldsymbol{\mu}, \boldsymbol{\Lambda})$, we obtain the tight ELBO presented in **Theorem** 4.1
 of the main paper:

$$\log \left( p(\mathbf{y}, \mathcal{U}|\mathbf{B}) \right) \geq L_1^*(\mathcal{U}, \mathbf{B}) = \frac{1}{2} \log |\mathbf{K}_{BB}| - \frac{1}{2} \log |\mathbf{K}_{BB} + \beta \mathbf{A}_1| - \frac{1}{2}\beta a_2 - \frac{1}{2}\beta a_3$$

$$+ \frac{\beta}{2} \mathrm{tr}(\mathbf{K}_{BB}^{-1}\mathbf{A}_1) - \frac{1}{2} \sum_{k=1}^{K} \|\mathbf{U}^{(k)}\|_F^2 + \frac{1}{2}\beta^2 \mathbf{a}_4^\top (\mathbf{K}_{BB} + \beta \mathbf{A}_1)^{-1}\mathbf{a}_4 + \frac{N}{2} \log(\frac{\beta}{2\pi}), \qquad (3)$$

 where $\| \cdot \|_F$ is Frobenius norm, and

$$\mathbf{A}_1 = \sum_j k(\mathbf{B}, \mathbf{x}_{\mathbf{i}_j}) k(\mathbf{x}_{\mathbf{i}_j}, \mathbf{B}), \quad a_2 = \sum_j y_{\mathbf{i}_j}^2, \quad a_3 = \sum_j k(\mathbf{x}_{\mathbf{i}_j}, \mathbf{x}_{\mathbf{i}_j}), \quad \mathbf{a}_4 = \sum_j k(\mathbf{B}, \mathbf{x}_{\mathbf{i}_j}) y_{\mathbf{i}_j}.$$

## 1.2 Binary Tensor

Next, let us look at the binary data. The case for binary tensors is more complex, because we have the additional variational posterior $q(\mathbf{z}) = \prod_j q(z_j)$. Furthermore, $q(\mathbf{v})$ and $q(\mathbf{z})$ are coupled in the original ELBO (see (2)). To eliminate $q(\mathbf{v})$ and $q(\mathbf{z})$, we use the following steps. We first fix $q(\mathbf{z})$, calculate the optimal $q(\mathbf{v})$ and plug it into $L_2$ (this is similar to the continuous case) to obtain an intermediate bound,

$$
\hat{L}_2(q(\mathbf{z}), \mathcal{U}, \mathbf{B}) = \max_{q(\mathbf{v})} L_2(q(\mathbf{v}), q(\mathbf{z}), \mathcal{U}, \mathbf{B}) = \frac{1}{2}\log|\mathbf{K}_{BB}| - \frac{1}{2}\log|\mathbf{K}_{BB} + \mathbf{A}_1| - \frac{1}{2}\sum_j \langle z_j^2 \rangle
$$
$$
- \frac{1}{2}a_3 + \frac{1}{2}\mathrm{tr}(\mathbf{K}_{BB}^{-1}\mathbf{A}_1) - \frac{N}{2}\log(2\pi) + \frac{1}{2}(\mathbf{K}_{BS}\langle\mathbf{z}\rangle)^\top(\mathbf{K}_{BB} + \mathbf{A}_1)^{-1})(\mathbf{K}_{BS}\langle\mathbf{z}\rangle)
$$
$$
+ \sum_j \int q(z_j)\log(\frac{p(y_{\mathbf{i}_j}|z_j)}{q(z_j)})\mathrm{d}z_j - \frac{1}{2}\sum_{k=1}^K \|\mathbf{U}^{(k)}\|_F^2 \tag{4}
$$

where $\langle\cdot\rangle$ denotes the expectation under the variational posteriors. Note that $\hat{L}_2$ has a similar form to $L_1^*$ in (3).

Now we consider to calculate the optimal $q(\mathbf{z})$ for $\hat{L}_2$. To this end, we calculate the functional derivative of $\hat{L}_2$ with respect to each $q(z_j)$:

$$
\frac{\delta\hat{L}_2}{\delta q(z_j)} = \log\frac{p(y_{\mathbf{i}_j}|z_j)}{q(z_j)} - 1 - \frac{1}{2}z_j^2 + c_{jj}\langle z_j\rangle z_j + \sum_{t\neq j} c_{tj}\langle z_t\rangle z_j.
$$

where $c_{tj} = k(\mathbf{x}_{\mathbf{i}_t}, \mathbf{B})(\mathbf{K}_{BB} + \mathbf{A}_1)^{-1}k(\mathbf{B}, \mathbf{x}_{\mathbf{i}_j})$ and $p(y_{\mathbf{i}_j}|z_j) = \mathbb{1}((2y_{\mathbf{i}_j} - 1)z_j \geq 0)$.

Solving $\frac{\delta\hat{L}_2}{\delta q(z_j)}$ being 0 with Lagrange multipliers, we find that the optimal $q(z_j)$ is a truncated Gaussian,

$$
q^*(z_j) \propto \mathcal{N}(z_j|c_{jj}\langle z_j\rangle + \sum_{t\neq j} c_{tj}\langle z_t\rangle, 1)\mathbb{1}((2y_{\mathbf{i}_j} - 1)z_j \geq 0).
$$

This expression is unfortunately not analytical. Even if we can explicitly update each $q(z_j)$, the updating will depend on all the other variational posteriors $\{q(z_t)\}_{t\neq j}$, making distributed calculation very difficult. This arises from the quadratic term $\frac{1}{2}(\mathbf{K}_{BS}\langle\mathbf{z}\rangle)^\top(\mathbf{K}_{BB} + \mathbf{A}_1)^{-1}(\mathbf{K}_{BS}\langle\mathbf{z}\rangle)$ in (4), which couples all $\{\langle z_j\rangle\}_j$.

To resolve this issue, we introduce an extra variational parameter $\boldsymbol{\lambda}$ to decouple the dependencies between $\{\langle z_j\rangle\}_j$ using the following lemma.

**Lemma 1.1.** *For any symmetric positive definite matrix* $\mathbf{E}$,

$$
\boldsymbol{\eta}^\top\mathbf{E}^{-1}\boldsymbol{\eta} \geq 2\boldsymbol{\lambda}^\top\boldsymbol{\eta} - \boldsymbol{\lambda}^\top\mathbf{E}\boldsymbol{\lambda}. \tag{5}
$$

*The equality is achieved when* $\boldsymbol{\lambda} = \mathbf{E}^{-1}\boldsymbol{\eta}$.

*Proof.* Define the function $f(\boldsymbol{\eta}) = \boldsymbol{\eta}^\top\mathbf{E}^{-1}\boldsymbol{\eta}$ and it is easy to see that $f(\boldsymbol{\eta})$ is convex because $\mathbf{E}^{-1} \succ 0$. Then using the convex conjugate, we have $f(\boldsymbol{\eta}) \geq \boldsymbol{\lambda}^\top\boldsymbol{\eta} - g(\boldsymbol{\lambda})$ and $g(\boldsymbol{\lambda}) \geq \boldsymbol{\eta}^\top\boldsymbol{\lambda} - f(\boldsymbol{\eta})$. Then by maximizing $\boldsymbol{\eta}^\top\boldsymbol{\lambda} - f(\boldsymbol{\eta})$, we can obtain $g(\boldsymbol{\lambda}) = \frac{1}{4}\boldsymbol{\lambda}^\top\mathbf{E}\boldsymbol{\lambda}$. Thus, $f(\boldsymbol{\eta}) \geq \boldsymbol{\lambda}^\top\boldsymbol{\eta} - \frac{1}{4}\boldsymbol{\lambda}^\top\mathbf{E}\boldsymbol{\lambda}$. Since $\boldsymbol{\lambda}$ is a free parameter, we can use $2\boldsymbol{\lambda}$ to replace $\boldsymbol{\lambda}$ and obtain the inequality (5). Further, we can verify that when $\boldsymbol{\lambda} = \mathbf{E}^{-1}\boldsymbol{\eta}$ the equality is achieved. $\square$

We now apply the inequality on the term $\frac{1}{2}(\mathbf{K}_{BS}\langle\mathbf{z}\rangle)^\top(\mathbf{K}_{BB} + \mathbf{A}_1)^{-1}\mathbf{K}_{BS}\langle\mathbf{z}\rangle$ in (4). Note that the quadratic term regarding all $\{z_j\}$ now vanishes, and instead a linear term $\boldsymbol{\lambda}^\top\mathbf{K}_{BS}\langle\mathbf{z}\rangle$ is introduced so that these annoying dependencies between $\{z_j\}_j$ are eliminated. We therefore obtain a more

 friendly intermediate ELBO,

$$\tilde{L}_2(\mathcal{U}, \mathbf{B}, q(\mathbf{z}), \boldsymbol{\lambda}) = \frac{1}{2}\log|\mathbf{K}_{BB}| - \frac{1}{2}\log|\mathbf{K}_{BB} + \mathbf{A}_1| - \frac{1}{2}\sum_j \langle z_j^2 \rangle - \frac{1}{2}a_3 + \frac{1}{2}\mathrm{tr}(\mathbf{K}_{BB}^{-1}\mathbf{A}_1)$$
$$- \frac{N}{2}\log(2\pi) + \sum_j \boldsymbol{\lambda}^\top k(\mathbf{B}, \mathbf{x}_{\mathbf{i}_j})\langle z_j \rangle - \frac{1}{2}\boldsymbol{\lambda}^\top(\mathbf{K}_{BB} + \mathbf{A}_1)\boldsymbol{\lambda} + \sum_j \int q(z_j)\log(\frac{p(y_{\mathbf{i}_j}|z_j)}{q(z_j)})\mathrm{d}z_j$$
$$- \frac{1}{2}\sum_{k=1}^K \|\mathbf{U}^{(k)}\|_F^2. \tag{6}$$

55 The functional derivative with respect to $q(z_j)$ is then given by

$$\frac{\delta\tilde{L}_2}{\delta q(z_j)} = \log\frac{p(y_{\mathbf{i}_j}|z_j)}{q(z_j)} - 1 - \frac{1}{2}z_j^2 + \boldsymbol{\lambda}^\top k(\mathbf{B}, \mathbf{x}_{\mathbf{i}_j})z_j.$$

56 Now solving $\frac{\delta\tilde{L}_2}{\delta q(z_j)} = 0$, we see that the optimal variational posterior has an analytical form:

$$q^*(z_j) \propto \mathcal{N}(z_j|\boldsymbol{\lambda}^\top k(\mathbf{B}, x_{\mathbf{i}_j}), 1)\mathbb{1}\big((2y_{\mathbf{i}_j} - 1)z_j \geq 0\big).$$

57 Plugging each $q^*(z_j)$ into (6), we finally obtain the tight ELBO as presented in **Theorem** 4.2 of the
58 main paper:

$$\log\big(p(\mathbf{y}, \mathcal{U}|\mathbf{B})\big) \geq L_2^*(\mathcal{U}, \mathbf{B}, \boldsymbol{\lambda}) = \frac{1}{2}\log|\mathbf{K}_{BB}| - \frac{1}{2}\log|\mathbf{K}_{BB} + \mathbf{A}_1| - \frac{1}{2}a_3$$
$$+ \sum_j \log\big(\Phi((2y_{\mathbf{i}_j} - 1)\boldsymbol{\lambda}^\top k(\mathbf{B}, \mathbf{x}_{\mathbf{i}_j}))\big) - \frac{1}{2}\boldsymbol{\lambda}^\top\mathbf{K}_{BB}\boldsymbol{\lambda} + \frac{1}{2}\mathrm{tr}(\mathbf{K}_{BB}^{-1}\mathbf{A}_1) - \frac{1}{2}\sum_{k=1}^K \|\mathbf{U}^{(k)}\|_F^2. \tag{7}$$

## 2  Gradients of the Tight ELBO

60 In this section, we present how to calculate the gradients of the tight ELBOs in (3) and (7) with
61 respect to the latent factors $\mathcal{U}$, the inducing points $\mathbf{B}$ and the kernel parameters.

62 Let us first consider the tight ELBO for continuous data. Because $\mathcal{U}$, $\mathbf{B}$ and the kernel parameters are
63 all inside the terms involving the kernel functions, such as $\mathbf{K}_{BB}$ and $\mathbf{A}_1$, we calculate the gradients
64 with respect to these terms first and then use the chain rule to calculate the gradients with respect to
65 $\mathcal{U}$ and $\mathbf{B}$ and the kernel parameters. Specifically, we consider the derivatives with respect to $\mathbf{K}_{BB}$,
66 $\mathbf{A}_1$, $a_3$ and $\mathbf{a}_4$. Using matrix derivatives and algebras (Minka, 2000), we obtain

$$\mathrm{d}L_1^* = \frac{1}{2}\mathrm{tr}\big((\mathbf{K}_{BB}^{-1} - (\mathbf{K}_{BB} + \beta\mathbf{A}_1)^{-1})\mathrm{d}\mathbf{K}_{BB}\big) - \frac{\beta}{2}\mathrm{tr}\big((\mathbf{K}_{BB} + \beta\mathbf{A}_1)^{-1}\mathrm{d}\mathbf{A}_1\big)$$
$$- \frac{\beta}{2}\mathrm{d}a_3 - \frac{\beta}{2}\mathrm{tr}(\mathbf{K}_{BB}^{-1}\mathbf{A}_1\mathbf{K}_{BB}^{-1}\mathrm{d}\mathbf{K}_{BB}) + \beta^2\mathrm{tr}(\mathbf{a}_4^\top(\mathbf{K}_{BB} + \beta\mathbf{A}_1)^{-1}\mathrm{d}\mathbf{a}_4)$$
$$+ \frac{\beta}{2}\mathrm{tr}(\mathbf{K}_{BB}^{-1}\mathrm{d}\mathbf{A}_1) - \frac{1}{2}\beta^2\mathrm{tr}\big((\mathbf{K}_{BB} + \beta\mathbf{A}_1)^{-1}\mathbf{a}_4\mathbf{a}_4^\top(\mathbf{K}_{BB} + \beta\mathbf{A}_1)^{-1}\mathrm{d}\mathbf{K}_{BB}\big)$$
$$- \frac{1}{2}\beta^3\mathrm{tr}\big((\mathbf{K}_{BB} + \beta\mathbf{A}_1)^{-1}\mathbf{a}_4\mathbf{a}_4^\top(\mathbf{K}_{BB} + \beta\mathbf{A}_1)^{-1}\mathrm{d}\mathbf{A}_1\big). \tag{8}$$

67 Next, we calculate the derivatives $\mathrm{d}\mathbf{K}_{BB}$, $\mathrm{d}\mathbf{A}_1$, $\mathrm{d}a_3$ and $\mathrm{d}\mathbf{a}_4$, which depend on the specific kernel
68 function form used in the model. For example, if we use the linear kernel, $\mathrm{d}\mathbf{K}_{BB} = 2\mathbf{B}^\top\mathrm{d}\mathbf{B}$
69 and $\mathrm{d}\mathbf{A}_1 = \sum_{j=1}^N k(\mathbf{B}, \mathbf{x}_{\mathbf{i}_j})(\mathbf{x}_{\mathbf{i}_j}\mathrm{d}\mathbf{B}^\top + \mathrm{d}\mathbf{x}_{\mathbf{i}_j}\mathbf{B}^\top) + (\mathrm{d}\mathbf{B}\mathbf{x}_{\mathbf{i}_j}^\top + \mathbf{B}\mathrm{d}\mathbf{x}_{\mathbf{i}_j}^\top)k(\mathbf{x}_{\mathbf{i}_j}, \mathbf{B})$ where $\mathbf{x}_{\mathbf{i}_j} =$
70 $[\mathbf{u}_{i_{j1}}^{(1)}, \ldots, \mathbf{u}_{i_{jK}}^{(K)}]$. Note that because $\mathbf{A}_1$, $a_3$ and $\mathbf{a}_4$ all have additive structures which involve
71 individual tensor entry $\mathbf{i}_j$ $(1 \leq j \leq N)$ and the major computation of the derivatives in (8) also
72 involve similar summations, the computation of the final gradients with respect to $\mathcal{U}$ and $\mathbf{B}$ and the
73 kernel parameters can easily be performed in parallel.

The gradient calculation for the tight ELBOs for binary tensors is very similar to the continuous case. Specifically, we obtain

$$
\begin{aligned}
\mathrm{d}L_2^* = {} & \frac{1}{2}\mathrm{tr}\big(\mathbf{K}_{BB}^{-1} - (\mathbf{K}_{BB} + \mathbf{A}_1)^{-1}\mathrm{d}\mathbf{K}_{BB}\big) - \frac{1}{2}\mathrm{tr}\big((\mathbf{K}_{BB} + \mathbf{A}_1)^{-1}\mathrm{d}\mathbf{A}_1\big) \\
& - \frac{1}{2}\mathrm{d}a_3 - \frac{1}{2}\mathrm{tr}(\mathbf{K}_{BB}^{-1}\mathbf{A}_1\mathbf{K}_{BB}^{-1}\mathrm{d}\mathbf{K}_{BB}) + \frac{1}{2}\mathrm{tr}(\mathbf{K}_{BB}^{-1}\mathrm{d}\mathbf{A}_1) - \frac{1}{2}\mathrm{tr}(\boldsymbol{\lambda}\boldsymbol{\lambda}^\top\mathrm{d}\mathbf{K}_{BB}) \\
& + \sum_{j=1}^N (2y_{\mathbf{i}_j} - 1)\frac{\mathcal{N}\big(\boldsymbol{\lambda}^\top k(\mathbf{B}, \mathbf{x}_{\mathbf{i}_j})|0, 1\big)}{\Phi\big((2y_{\mathbf{i}_j} - 1)\boldsymbol{\lambda}^\top k(\mathbf{B}, \mathbf{x}_{\mathbf{i}_j})\big)}\boldsymbol{\lambda}^\top\mathrm{d}k(\mathbf{B}, \mathbf{x}_{\mathbf{i}_j}).
\end{aligned} \tag{9}
$$

We can then calculate the derivatives $\mathrm{d}\mathbf{K}_{BB}$, $\mathrm{d}\mathbf{A}_1$, $\mathrm{d}a_3$ and each $\mathrm{d}k(\mathbf{B}, \mathbf{x}_{\mathbf{i}_j})(1 \leq j \leq N)$ and then apply the chain rule to calculate the gradient with respect to $\mathcal{U}$, $\mathbf{B}$ and the kernel parameters.

## 3 Fixed Point Iteration for $\boldsymbol{\lambda}$

In this section, we give the convergence proof of the fixed point iteration of the variational parameters $\boldsymbol{\lambda}$ in the tight ELBO for binary tensors. While $\boldsymbol{\lambda}$ can be jointly optimized via gradient based approaches with $\mathcal{U}$, $\mathbf{B}$ and the kernel parameters, we empirically find that combining this fixed point iteration can converge much faster. The fixed point iteration is given by

$$
\boldsymbol{\lambda}^{(t+1)} = (\mathbf{K}_{BB} + \mathbf{A}_1)^{-1}(\mathbf{A}_1\boldsymbol{\lambda}^{(t)} + \mathbf{a}_5) \tag{10}
$$

where

$$
\mathbf{A}_1 = \sum_j k(\mathbf{B}, \mathbf{x}_{\mathbf{i}_j})k(\mathbf{x}_{\mathbf{i}_j}, \mathbf{B}), \quad \mathbf{a}_5 = \sum_j k(\mathbf{B}, \mathbf{x}_{\mathbf{i}_j})(2y_{\mathbf{i}_j} - 1)\frac{\mathcal{N}\big(k(\mathbf{B}, \mathbf{x}_{\mathbf{i}_j})^\top\boldsymbol{\lambda}^{(t)}|0, 1\big)}{\Phi\big((2y_{\mathbf{i}_j} - 1)k(\mathbf{B}, \mathbf{x}_{\mathbf{i}_j})^\top\boldsymbol{\lambda}^{(t)}\big)}.
$$

We now show that the fixed point iteration not only always converges, but also improves the ELBO in (7) after every update of $\boldsymbol{\lambda}$ (see **Lemma** 4.3 in the main paper).

Specifically, given $\mathcal{U}$ and $\mathbf{B}$, from Section 1 we have

$$
L_2^*\big(\boldsymbol{\lambda}^{(t)}\big) = \max_{q(\mathbf{z})} \tilde{L}_2\big(\boldsymbol{\lambda}^{(t)}, q(\mathbf{z})\big) = \tilde{L}_2\big(\boldsymbol{\lambda}^{(t)}, q_{\boldsymbol{\lambda}^{(t)}}(\mathbf{z})\big)
$$

where $q_{\boldsymbol{\lambda}^{(t)}}(\mathbf{z})$ is the optimal variational posterior: $q_{\boldsymbol{\lambda}^{(t)}}(\mathbf{z}) = \prod_j q_{\boldsymbol{\lambda}^{(t)}}(z_j)$ and $q_{\boldsymbol{\lambda}^{(t)}}(z_j) \propto \mathcal{N}\big(z_j|k(\mathbf{B}, \mathbf{x}_{\mathbf{i}_j})^\top\boldsymbol{\lambda}^{(t)}, 1\big)\mathbb{1}\big((2y_{\mathbf{i}_j} - 1)z_j \geq 0\big)$.

Now let us fix $q_{\boldsymbol{\lambda}^{(t)}}(\mathbf{z})$ and derive the optimal $\boldsymbol{\lambda}$ by solving $\frac{\partial \tilde{L}_2}{\partial \boldsymbol{\lambda}} = 0$. We then obtain the update of $\boldsymbol{\lambda}$: $\boldsymbol{\lambda}^{(t+1)} = (\mathbf{K}_{BB} + \mathbf{A}_1)^{-1}\big(\sum_j k(\mathbf{B}, \mathbf{x}_{\mathbf{i}_j})\langle z_j\rangle\big)$ where $\langle z_j\rangle$ is the expectation of the optimal variational posterior of $z_j$ given $\boldsymbol{\lambda}^{(t)}$, i.e., $q_{\boldsymbol{\lambda}^{(t)}}(z_j)$. Obviously, we have

$$
\tilde{L}_2\big(\boldsymbol{\lambda}^{(t)}, q_{\boldsymbol{\lambda}^{(t)}}(\mathbf{z})\big) \leq \tilde{L}_2\big(\boldsymbol{\lambda}^{(t+1)}, q_{\boldsymbol{\lambda}^{(t)}}(\mathbf{z})\big).
$$

Further, because $L_2^*(\boldsymbol{\lambda}^{(t)}) = \tilde{L}_2\big(\boldsymbol{\lambda}^{(t)}, q_{\boldsymbol{\lambda}^{(t)}}(\mathbf{z})\big)$ and

$$
\tilde{L}_2\big(\boldsymbol{\lambda}^{(t+1)}, q_{\boldsymbol{\lambda}^{(t)}}(\mathbf{z})\big) \leq \tilde{L}_2\big(\boldsymbol{\lambda}^{(t+1)}, q_{\boldsymbol{\lambda}^{(t+1)}}(\mathbf{z})\big) = L_2^*(\boldsymbol{\lambda}^{(t+1)})
$$

we conclude that $L_2^*(\boldsymbol{\lambda}^{(t)}) \leq L_2^*(\boldsymbol{\lambda}^{(t+1)})$. Now, we plug the fact that $\langle z_j\rangle = w_j^{(t)} + k(\mathbf{B}, \mathbf{x}_{\mathbf{i}_j})^\top\boldsymbol{\lambda}^{(t)}$ given $q_{\boldsymbol{\lambda}^{(t)}}(z_j)$ into the calculation of $\boldsymbol{\lambda}^{(t+1)}$, merge and arrange the terms. We then obtain the fixed point iteration for $\boldsymbol{\lambda}$ in (10). Finally since $L_2^*$ is upper bounded by the log model evidence, the fixed point iteration always converges.

## 4 Application on Click-Through-Rate Prediction

In this section, we report the results of applying our nonlinear tensor factorization approach on Click-Through-Rate (CTR) prediction for online advertising.

We used the online ads click log from a major Internet company, from which we extracted a four mode tensor (*user*, *advertisement*, *publisher*, *page-section*). We used the first three days's log on May 2015,

trained our model on one day's data and used it to predict the click behaviour on the next day. The sizes of the extracted tensors for the three days are $179K \times 81K \times 35 \times 355$, $167K \times 78K \times 35 \times 354$ and $213K \times 82K \times 37 \times 354$ respectively. These tensors are very sparse ($2.7 \times 10^{-8}\%$ nonzeros on average); in other words, the observed clicks are very rare. However, we do not want our prediction completely bias toward zero (i.e., non-click); otherwise, ads ranking and recommendation will be infeasible. Thus we sampled non-clicks of the same quantity as the clicks for training and testing. Note that training CTR prediction models with comparable clicks and non-click samples is common in online advertising systems (Agarwal et al., 2014). The number of training and testing entries used for the three days are $(109K, 99K)$, $(91K, 103K)$ and $(109K, 136K)$ respectively.

We compared with popular methods for CTR prediction, including logistic regression and linear SVM, where each tensor entry is represented by a set of binary features according to the indices of each mode in the entry. We used the distributed implementations in spark MLlib.

The results are reported in Table 1, in terms of AUC. It shows that our model improves logistic regression and linear SVM by a large margin, on average 20.7% and 20.8% respectively. Therefore, although we have not incorporated side features, such as user profiles and advertisement attributes, our tentative experiments have shown a promising potential of our model on CTR prediction task.

Table 1: CTR prediction accuracy on the first three days of May 2015. "1-2" means using May 1st's data for training and May 2nd's data for testing; similar are "2-3" and "3-4".

| Method | 1-2 | 2-3 | 3-4 |
|---|---|---|---|
| Logistic regression | 0.7360 | 0.7337 | 0.7538 |
| Linear SVM | 0.7414 | 0.7332 | 0.7540 |
| Our model | **0.8925** | **0.8903** | **0.9054** |