[Reviews · NeurIPS 2016]

Reviewer 1

Summary

The paper proposes a non-linear tensor factorization approach, using a Gaussian Process to map from the (concatenated) latent factors to the observations (continuous or binary). The paper presents an efficient inference procedure based on sparse GP variational inference.

Qualitative Assessment

Interesting paper where the main merit is what appears to be a very efficient inference procedure / model formulation for a nonlinear tensor factorization. The experimental evaluation is fairly convincing, and the approach seems promising. Because of how the GP is used in the model, I would have been interested in some more discussion of the influcence of different choices of kernels.

Confidence in this Review

2-Confident (read it all; understood it all reasonably well)


Reviewer 2

Summary

The paper presents a distributed nonlinear tensor factorization model, that reduces costs by essentially devising a sparsity-inducing mechanism. In addition, they perform variational inference by devising an evidence lower bound expression that enables highly decoupled, parallel computations and high-quality inference; they use the MAPREDUCE framework to this end.

Qualitative Assessment

The method is both novel and interesting. The theoretical grounding of the method is solid, while the provided insights into the properties of the method are satisfactory. The experimental evaluation is in general convincing. An aspect that certainly needs more elaboration upon is how the number of latent factors and GP inducing points turns out to affect model performance (both modeling/predictive capacity as well as computational costs).

Confidence in this Review

2-Confident (read it all; understood it all reasonably well)


Reviewer 3

Summary

This paper places a Gaussian process prior over the mapping from latent factors of modes in a tensor to their observed entries. This allows interactions to be captured in latent space between modes using a kernel function. They derive a tractable ELBO (using inducing points) to perform variational inference on this model, and cleverly design the ELBO to enable parallelized computation using MapReduce. Indeed, they demonstrate their methods scales to massive real-world datasets.

Qualitative Assessment

This paper was a pleasure to read. Their model is simple, but parallelizing the inference is not, and they developed new tricks to write down a tractable ELBO for variational inference decomposes appropriately. The experiment section was amazing and had everything I look for: iteration times, massive real-world datasets, mapreduce? I couldn't ask for more - thank you. However, I would have appreciated more intuition and clarification on why concatenation of latent factors makes sense as the input data. This is a strange operation and seems open to further research: why not take the element-wise product? It seems the choice of concatenation is important and thus should be clarified. Maybe the operation of concatenation (or addition, element-wise product, cosine distance, etc) can be designed according to the data?

Confidence in this Review

2-Confident (read it all; understood it all reasonably well)


Reviewer 4

Summary

This paper proposes a tensor factorization algorithm that can be viewed as a tensor completion problem, in which a ``low-rank'' decomposition of the tensor is constructed using partial observations of the tensor. It is based on the Tucker decomposition with appropriate priors for the Tucker factors $U$. It considers both binary and continuous tensors. For continuous tensors the formulation is based on the model of equation 2, where the kernelized prior is based defined in lines 113--120. Furthermore because the covariance matrix $K$ that appears in EQ 2 is very large, an approximation by using the scheme in 18. A fixed-step gradient descent scheme is used (combined with L-BFGS). Additional optimizations are discussed for use on a distributed architecture using the MapReduce and the SPARK system. The paper concludes with numerical experiments that demonstrate the performance of the algorithm in terms of accuracy of the approximation and AUC values for several datasets.

Qualitative Assessment

REVIEW The results in Figure 2 look quite good, but why? Is it because kernelization or is it because of the simpler covariance structure? Overall no explanation on why the method performs better is given. Timing comparisons with the other factorizations are not presented, so is the method faster? If not, what happens if we increase the ranks for the other methods. If there are so many fudging factors, where the other methods similarly optimized? Not sure that the bounds are useful. How are they used in the algorithm? There are numerous parameters: prior for W core tensor, kernel form and kernel parameters, $\beta$ for observations, the number of inducing points, regularization for $U$ (first term in Eq 2), the size $p$ for the B matrix. This seems an almost impossible method to use, how do you choose all these parameters? This formulation introduces more non-convexity with having to solve for the inducing points B. Also, how is the algorithm terminated? In the experimental session the splitting between testing and non-testing is arbitrary. SECONDARY COMMENTS Are the inducing points solved for or are there a subset of the original ones? How exactly is EQ 2 combined with the equation after line 148? Question on Line 38: ``First, it can capture highly nonlinear interactions in the tensor, and is flexible enough to incorporate arbitrary subset of (meaningful) tensorial entries for the training.'' How is this different from other tensor factorizations? What does ``Highly nonlinear'' mean? How Gaussian priors allow more nonlinear interactions? These statements seem quite ad hoc to me. Line 45: ``For example, one can choose a combination of balanced zero and nonzero elements to overcome the learning bias.'' But in the first paragraph of the introduction it is mentioned that zero elements are meaningless. Not sure what this sentence means. The algorithmic complexity in 4.4 does not involve the number of iterations. Also, how do you invert K_BB, that should have at least p^3 complexity, right? I guess you just ignore this term? There is way too much material in this paper. If it ends up resubmitted, I recommend that first the focus is on the formulation and detailed numerical experiments, and then on the distributed runs. The comparison with linear logistic and svm is interesting, but one should compare with kernelized versions of those methods.

Confidence in this Review

1-Less confident (might not have understood significant parts)


Reviewer 5

Summary

The paper proposes a distributed nonlinear tensor factorization for prediction.. The proposed method targets scenarios where there are many missing entries in the data. It is applicable to CTR prediction. The authors develop a tight variational evidence lower bound that helps them develop the distributed implementation of the method.

Qualitative Assessment

The paper is well written and well structured. The authors have performed small and large scale experiments to validate their proposed non-linear tensor factorization. The authors apply a nonlinear function to the corresponding latent factors in Section 3, which results in the full covariance matrix to be disposed of a the Kronecker product structure. They then argue that since this holds they can choose any subset of the entries for training. I think this statement needs clarification. Can you provide more detail about why this is true and we can choose any subset to train? Following this argument the authors use a balanced number of zero and nonzero entries for training. However, the unbiasedness of the zeros and non-zeros in the original data is indicative of the data behavior. If you balance them and then do training, why should the results still preserve the statistics of the original data? Minor comments: --Avoid using TGP in the abstract before defining what it stands for. --Typo in line 64: achieve a significant ...

Confidence in this Review

2-Confident (read it all; understood it all reasonably well)